# OpenReview forum: "When Ethics and Payoffs Diverge: LLM Agents in Morally Charged Social Dilemmas"
_ICLR.cc/2026/Conference — Submitted to ICLR 2026_

### Official Review · Reviewer_w88c · 2025-10-31

**Soundness:** 2
**Presentation:** 3
**Contribution:** 1
**Rating:** 2
**Confidence:** 5

**Summary:**

This work studies how LLMs behave when moral obligations conflict with incentives in social dilemmas. The authors present a simulation framework called MoralSim that essentially is a collection of situations that reflect social dilemmas expressed in natural language. The authors evaluate several LLMs considering Prisoner's Dilemma and Public Good games. The findings show that models vary widely in their tendency to act morally and lack consistent ethical behavior across situations, suggesting that LLMs may be unreliable for decision-making roles requiring alignment with ethical norms.

**General note**: I reviewed this manuscript for a previous conference. I would like to note that I went through the paper carefully, noting the main differences. The core contributions are essentially the same apart from the causality analysis; hence, I find myself in a position to repeat my critiques. Please note that the list of weaknesses below takes into consideration the revised version of the paper with respect to my initial review. I would also like to note that my comment about the naming of the “morality score” was addressed, and in this version, the authors talk about “cooperation score”. I noted the following changes with respect to the version I reviewed below (these are the main ones I was able to clearly identify):

- The morality score has been renamed cooperation score.
- A section about causal effect estimation has been added (plus a discussion of the treatment effects later in the paepr).
- The discussion of the results regarding Q3 and Q4 has been extended.
- There is a new section about reasoning trace analysis in the appendix.

These points/observations did not affect my review. I treated this paper completely *as new*. I believe that these are independent conferences and we should consider the manuscripts in their own right each time. I added this note explicitly in order to avoid potential criticisms about the fact I raised again some of the concerns of my previous review (since those parts are unchanged).

**Strengths:**

- This is indeed a very interesting topic.
- The paper is well written and very easy to follow. The evaluation of the work is conducted.

**Weaknesses:**

- This is a well-written paper, but it is difficult to identify a substantial contribution with respect to the state of the art. In fact, the papers listed under “LLMs in Game Theory Settings” focus directly or indirectly on decision-making problems in social dilemmas. In fact, the authors do not really consider actual moral frameworks in their analysis as these works do.  The authors claim that various prior research has already explored LLMs’ moral reasoning and strategic behaviour separately. The authors of [Tennant et al., 2024] (actually published and presented at ICLR 2025) essentially not only present a variety of games (including the Iterated Prisoner's Dilemma) but also study how to run a fine-tuning procedure to examine the effects of moral decision-making.
- The authors present very limited analysis in terms of sensitivity to variations of the prompts, including the agent setup that is discussed in Section 3.3 (e.g., descriptions of the setting, personal memory, and current task). These might have substantial effects and should be discussed by the authors in my opinion.
- It is quite surprising that the authors considered moral dilemmas, but the actual dynamics of the responses are only partially analysed/considered by the authors. In fact, the authors mainly focus on the choice of the single agents. The authors also consider the opponent alignment, but this is quite confusing since it appears to the reviewer quite orthogonal to the problem of acting morally. The authors consider the relative payoff, but considering the fact that these are classic (repeated) games, the analysis of the actual cumulative payoff would have probably been more informative from a game theory point of view.
- The survival rate score is interesting, but it appears rather disjointed considering the core topic of the paper. With respect to the statistical validity of the simulations, it is unclear how different repetitions of the games have been implemented.
- The paper essentially lacks a related work section. The authors moved it to the Appendix, but that is outside the 10 pages of the main body of the paper. It seems to me a way for going above the 10 page limit: this is not fair towards the other ICLR authors in my opinion.

**Questions:**

I do not have specific questions for the authors. My core concern is about the very limited contribution of this work with respect to the state of the art.

---

> ### Author Response · Authors · 2025-12-02
>
> **Contribution and Relation to Prior Work**
>
> We appreciate the reviewer’s careful reading and the note that a previous version of this work was reviewed earlier. We agree that MoralSim sits in a rapidly growing space of “LLMs in game-theoretic settings”; we revised the paper to state much more clearly where our contribution lies.
>
> **Tennant et al.** propose using intrinsic moral rewards to train a single LLM-based agent in the Iterated Prisoner’s Dilemma and a small set of matrix games. Their main question is: can an agent be fine-tuned so that it learns and generalizes moral strategies across games? The contributions centre on reward design and fine-tuning. By contrast, **MoralSim** is designed as a controlled evaluation benchmark for already-trained frontier models, not a training method. Our focus is on how nine off-the-shelf models behave without additional fine-tuning when faced with explicit conflicts between moral norms and payoff. This is particularly relevant in practice because the most capable and widely deployed models (e.g., GPT, Claude, Gemini) are typically closed-source and used directly in their provider-aligned form. MoralSim therefore characterizes the behavior that can be expected under current deployment practices, rather than under specific post-training regimes.
>
> Conceptually, Tennant et al. answer “can we train agents to be moral in and around IPD?”, while MoralSim asks “do the most capable models available today behave morally in a context-robust way across different games and moral framings?”. Our main conceptual contribution is to show that they do not: none of the tested frontier models exhibits a stable “moral policy”. The same model can behave predominantly morally in one combination of game and framing, and predominantly self-interested in another, purely due to changes in game structure, moral domain, opponent strategy, or survival risk.
> (We discuss other related benchmarks, such as MACHIAVELLI, GovSim, and Survival Games, in response to reviewer VVKc.)
>
> **Sensitivity and Dynamics**
>
> We fully agree with the reviewer that scientific robustness requires analysing sensitivity to prompt and setup variations. In the current submission, we already (i) run multiple seeds per configuration at temperature 0, and (ii) include a paraphrase robustness study in which we rephrase the scenarios and observe only small (~2 percentage point) deviations in cooperation and payoff metrics for GPT-4o and DeepSeek-R1. In the revision, we will surface these robustness results earlier in the main text, rather than leaving them to later sections. This is intended to address directly the concern that small prompt or setup changes might fundamentally alter our conclusions.
>
> Regarding dynamics and metrics, we also appreciate the concern that our analysis could appear to focus mainly on single-step choices. In fact, our relative payoff metric is cumulative over the entire trajectory of the repeated game. Survival, rather than acting as auxiliary score, acts as a situational factor in the dilemma: we explicitly manipulate survival risk as a treatment, and we find that, for several models, it induces substantial drops in cooperation (e.g., −17.6pp for Claude-3.7-Sonnet, −24.4pp for o3-mini as seen in Table 2 of the manuscript). Together, cooperation, payoff, and survival define the trade-off surface we study.

---

### Official Review · Reviewer_mAm7 · 2025-11-01

**Soundness:** 3
**Presentation:** 4
**Contribution:** 2
**Rating:** 6
**Confidence:** 5

**Summary:**

The paper presents MORALSIM, a new framework for evaluating the moral behavior of LLM agents in situations where ethical norms conflict with personal interests and benefits. The authors investigate the behavior of 9 modern language models (including Claude-3.7-Sonnet, GPT-4o, Deepseek-R1, o3-mini, and others) in two classic game scenarios: Prisoner's Dilemma and the Public Goods Game. Each game is embedded in three different moral contexts: Contractual Reporting, Privacy Protection, and Green Production. The full factorial design of the experiment is used, varying the type of game, the moral context, the opponent's behavior (always cooperating / always betraying) and the risk of survival. The key result: none of the tested models demonstrates consistent moral behavior in all scenarios. The proportion of morally oriented actions varies from 7.9% (Qwen-3) to 76.3% (GPT-4o-mini). The authors use causal analysis (Average Treatment Effects) to determine the factors influencing moral decisions, and show that the structure of the game, the specific moral context, and the opponent's behavior have the greatest impact.

**Strengths:**

- A full-factor design with clear manipulations (type of game, moral context, survival risk, opponent behavior) allows you to isolate the effects of each factor
- Quality of analysis. Analysis of ~3500 reflections of agents reveals decision-making mechanisms. Causal assessment through ATEs yields quantitative effects with confidence intervals. It is shown that profit maximizer models (Deepseek-R1, Qwen) rely on profit maximization, while more cooperative ones (Claude, GPT-4o) more often take into account moral considerations.
- All models show a decrease in morale precisely when the user is most vulnerable (at risk of bankruptcy), which raises important issues of AI security.
- The code is open, the prompts are documented in detail.

**Weaknesses:**

1. PD and PGG only. Other structures (for example, Trust Game, Stag Hunt) could reveal other patterns of moral behavior. An extension to asymmetric games would be especially valuable.
2. In the real world, agents can often negotiate, which significantly changes the dynamics of cooperation. The authors acknowledge this, but do not investigate it.
3. For some models (Claude-3.7-Sonnet, Gemini-2.5-Flash), versions without reasoning mode were used for cost reasons. Given that the analysis has shown the importance of reasoning, this limits the conclusions.
4. Multi-agent scenarios (N>2) could better reflect social dynamics and collective responsibility.

**Questions:**

1. You have shown invariance to paraphrases, but how sensitive are the results to more fundamental changes in the presentation of the problem? For example, what if we present the same dilemmas through different metaphors or change the order in which options are presented?
2. Does the moral behavior of agents change as they gain experience in repetitive games? Are there signs of "moral learning" or adaptation of strategies over time?
3. Moral norms vary between cultures. Do you plan to investigate how different cultural contexts affect the moral behavior of LLM agents?

---

> ### Author Response · Authors · 2025-11-20
>
> **Weaknesses (Game structures, Negotiation, N>2)**
>
> We thank the reviewer for the detailed and constructive assessment, especially the recognition of our full-factorial design and analysis quality. We fully agree that PD and PGG do not exhaust the space of social dilemmas. Our choice of these two canonical games was driven by the aim to study moral vs payoff trade-offs in two structurally distinct but analytically tractable settings, while keeping the design small enough to run a full factorial over game, moral context, opponent behaviour, and survival risk for nine frontier models. In the revision, we will make this more explicit and highlight that MoralSim is designed to be extensible to other game structures and to N>2 multi-agent scenarios.
>
> We also agree that negotiation and richer communication can significantly change cooperation dynamics. We deliberately focus on non-communicating agents to isolate the effects of game structure and moral framing. We will clarify this design choice and explicitly discuss a communication/negotiation phase in the Limitations sections.
>
> **Questions: Robustness, Dynamics, and Cultural Norms**
>
> On robustness and dynamics, we currently show robustness to paraphrases of the scenarios (Sec 3.5), but we agree it is also valuable to examine more fundamental variations. In the revision, we will make this analysis more prominent and add a more fine-grained to reinforce that our conclusions are not driven by small wording variations (See response to reviewer VVKc).
>
> To address the question about “moral learning” or adaptation over repeated games, we analyzed cooperation round-by-round using the existing trajectories. For each model and treatment condition (game, moral context, opponent, survival), we computed early vs late cooperation (first half vs second half of rounds) and a per-round linear trend. Across all conditions with valid trajectories, median $|\Delta\text{cooperation}|$ between early and late is 3.3 percentage points; the distribution is mildly biased towards decay (mean $\Delta \approx -6.3$ pp).
>
> These effects are systematically stronger in harsher settings (defecting opponents, survival risk, moral framings where cooperation is costly) and near-flat in the baseline context and under cooperative opponents. In LLM-vs-LLM two-player runs, we observe a similar pattern: in the Prisoner’s Dilemma, cooperation remains essentially stable or slightly increases, whereas in the Public Goods Game it decreases by $\approx$ 12–24 pp over the episode, consistent with free-riding.
>
> Overall, this suggests that the dominant differences in behaviour arise between experimental factors (game structure, moral framing, opponent, survival), rather than from strong within-condition adaptation or unstable round-specific artefacts.
>
> **Table 1: Early–late cooperation changes (late − early) across conditions.**
>
> | Condition subset | #conds | mean Δcoop (pp) | median Δcoop (pp) | median abs Δcoop (pp) |
> | :-- | --: | --: | --: | --: |
> | **All conditions** | 287 | -6.3 | -3.3 | 3.3 |
> | **PD** | 146 | -4.5 | -3.3 | 3.3 |
> | **PGG** | 146 | -8.2 | 0.0 | 3.3 |
> | **Opponent: cooperate** | 146 | -4.4 | 0.0 | 0.0 |
> | **Opponent: defect** | 146 | -8.3 | -6.7 | 6.7 |
> | **Survival: off** | 146 | -5.4 | -3.3 | 3.3 |
> | **Survival: on** | 146 | -7.3 | -3.3 | 3.3 |
> | **Baseline (no moral ctx)** | 72 | -1.2 | 0.0 | 0.0 |
> | **Moral contexts (all)** | 220 | -8.0 | -3.3 | 6.7 |
>
> **Table 2: Mean early–late change in cooperation per model and condition subset.**
>
> | Condition | Claude | DeepSeek-chat | DeepSeek-R1 | Gemini-2.5 | LLaMA-70B | Qwen3-235B | GPT-4o | o3-mini |
> | :------------------- | :----: | :-----------: | :---------: | :--------: | :-------: | :--------: | :----: | :-----: |
> | **All**   |  -8.0  |   -8.4  |    -4.3     |  -9.8    |   -13.4   |    -2.8    |  -2.3  |  -5.9   |
> | **PD**   |  -5.8  |     -6.0      |    -1.0     |    -6.7    |   -14.1   |    -1.7    |   2.6  |  -6.9   |
> | **PGG**  | -10.2  |    -10.8      |    -7.5     |   -12.9    |   -12.6   |    -4.0    |  -8.0  |  -4.9   |
> | **Opp: coop**  |  -2.1  |     -8.0   |    -2.5     |    -5.8    |   -11.7   |    -1.7    |  -5.0  |  -3.3   |
> | **Opp: defect** | -13.9  |     -8.8    |    -6.0     |   -13.7    |   -15.1   |    -4.0    |   0.7  |  -8.7   |
> | **Survival: off** |  -6.0  |     -6.0    |    -5.6     |    -6.0    |   -11.7   |  -4.6    |  -6.5  |  -4.0   |
> | **Survival: on** |  -9.9  |    -10.7  |    -2.9     |   -13.5    |   -15.1   |  -1.0    |   2.4  |  -8.0   |
> | **Baseline** |   6.7  |     -4.5      |    -0.8     |     0.4    |    -7.9   |     0.0    |  -5.0  |  -1.2   |
> | **Moral: non-baseline** | -12.9 |    -9.7     |   -5.4   |   -13.2  |   -15.2   |    -3.7    |  -1.4  |  -7.5   |
>
> Finally, we agree that moral norms vary across cultures. MoralSim in its current form encodes a single, broadly Western normative perspective (e.g., on contracts, privacy, and sustainability). We will mention this in our the Limitations sections.

---

> > ### Comment · Reviewer_mAm7 · 2025-11-25
> >
> > Thanks for the clarifications — they were helpful and addressed my concerns. I’ll retain my score, and I’m still convinced that my relatively high evaluation accurately reflects the quality of the work.

---

### Official Review · Reviewer_8FDD · 2025-11-01

**Soundness:** 2
**Presentation:** 3
**Contribution:** 3
**Rating:** 6
**Confidence:** 4

**Summary:**

The authors introduce MoralSim, a benchmark that focuses on studying LLM behaviors in scenarios where explicitly forced to trade-off between moral behavior and rewards. The authors encase these scenarios in realistic settings to provide greater fidelity to underlying LLM behavior and perform analysis of different behavioral questions in LLMs to find that

**Strengths:**

- The paper is well-written and the experiments are well-designed.
- The settings developed by the paper more clearly expose trade-offs in moral behavior compared to prior work.
- There are clear behavioral takeaways from the paper, e.g., opponent behavior meaningfully steers LLM actions or that moral context improves the morality of LLM behaviors.

**Weaknesses:**

While the games offer a step towards realism, they still do not capture the full nuances of reality. In particular:
- All the games are two-player. Many real-world settings involve multiple players with different levels of power and interlocking incentives.
- The text of the games themselves is still somewhat unrealistic. There is an explicit payoff structure described in the system prompt (e.g., Figure 2), whereas in the real world an LLM agent would need to uncover those trade-offs themselves.
However, I realize that these nuances are quite tricky to incorporate and would make some of the analysis less clean, so I don't think that should hold the paper back.

There is no discussion or analysis of whether or not the LLMs participating in the games recognize that they are in the game. Recent research into scheming (https://www.antischeming.ai/) suggests that frontier LLMs may recognize that they are being evaluated, which could question the validity of the research results. However, it is difficult to assess scheming without access to the full reasoning trace, so I also don't count this as a strong weakness.

**Questions:**

- Would it be possible at all to analyze whether the LLMs recognize that they are playing a game and if that would alter their behavior in any sense? Would it be possible to attempt to induce "evaluation-awareness" into the LLMs (e.g., by being more suggestive with the wording that the environment is a game) and see how that affects the results?

---

> ### Author Response · Authors · 2025-11-20
>
> **Realism and Game Design**
>
> We appreciate the reviewer’s positive assessment of the writing, design, and clarity of behavioral takeaways. We agree that our environments may not always capture the full nuance of real-world settings. This simplification is deliberate: our aim is to complement simulations such as GovSim and Survival Games with a benchmark where we can precisely control game structure, moral framing, opponent behaviour, and survival risk, and then use causal ATEs to disentangle their marginal effects. We will clarify this realism vs identifiability design choice more explicitly in the Introduction and Limitations, and emphasize that MoralSim is intended as a diagnostic layer rather than a full simulation of deployment settings.
>
> **Evaluation-Awareness and Scheming**
>
> On evaluation-awareness and potential scheming, we agree this is an important emerging concern. In our current setup, agents are always placed in-role (e.g., CEOs, auditors) and are never told they are being “evaluated” or “graded” in a benchmark. Inspired by recent work that annotates chain-of-thought for signs that models realize they are in an evaluation [1,2], we have now conducted am analysis on our own reflection logs using a separate LLM judge (GPT-5.1 high).
>
> The judge sees the (reconstructed) scenario description and is instructed to label a run as evaluation-aware only when the reflection explicitly breaks the fourth wall (e.g., “this is a test/experiment/simulation”, “the user is asking this question”). Across 1,280 runs (320 per model), the judge labels only 34 reflections (2.7%) as explicitly evaluation-aware, all from DeepSeek-R1:
>
> **Table 1: Explicit evaluation-awareness by model (LLM judge, strict criterion).**
>
> | Model | Reflections evaluated | Eval-aware reflections | Eval-aware rate |
> | :--- | :--- | :--- | :--- |
> | Claude-3.7-Sonnet | 320 | 0 | 0.0% |
> | DeepSeek-R1 | 320 | 34 | 10.6% |
> | Llama-3.3-70B-Instruct | 320 | 0 | 0.0% |
> | GPT-4o | 320 | 0 | 0.0% |
> | **Overall** | **1,280** | **34** | **2.7%** |
>
> For the 34 DeepSeek-R1 runs flagged as evaluation-aware, the judge most often flags cases where the model refers to an external problem/prompt or explicitly describes the situation as a test:
>
> **Table 2: Patterns in DeepSeek-R1 eval-aware reflections (34 runs).**
>
> | Pattern | Description (per judge) | % of DeepSeek eval-aware (non-mutually exclusive) |
> | :--- | :--- | :--- |
> | **external_problem_prompt** | Talks about “the problem / prompt / task” | 64.7% |
> | **user_assistant_external** | Refers to external “user” or “assistant” | 61.8% |
> | **tested_or_simulation** | Explicitly mentions being “tested” | 44.1% |
>
> Taken together, this suggests that under our in-role prompts, explicit evaluation-awareness is rare in MoralSim and concentrated in a single payoff-maximizing model. While this does not rule out more subtle forms of situational awareness, it makes it unlikely that our main cooperation and payoff patterns are strictly a result of performative compliance. We will incorporate this analysis into the revised paper and treat evaluation-awareness as an important axis for future work.
>
> [1] Schoen et al., *Stress Testing Deliberative Alignment for Anti-Scheming Training*, 2025.
>
> [2] Phuong et al., *Evaluating Frontier Models for Stealth and Situational Awareness*, 2025.

---

### Official Review · Reviewer_VVKc · 2025-11-02

**Soundness:** 3
**Presentation:** 2
**Contribution:** 2
**Rating:** 4
**Confidence:** 5

**Summary:**

The paper investigates the behavior of large language models (LLMs) as agents in social dilemmas that include moral constraints. Specifically, it explores how these models make decisions when moral principles conflict with strategies that yield higher rewards. To address this, the authors propose a new evaluation framework called MoralSim (Moral Behavior Social Dilemma Simulation), which integrates classic game-theoretic environments (such as repeated Prisoner's Dilemma and Public Goods Game) with real-world moral scenarios.

**Strengths:**

- The MoralSim framework combines classic game-theoretic scenarios with real-world moral dilemmas, enabling a systematic and comprehensive evaluation. This framework is more structured than previous fragmented tests. For example, compared to the MACHIAVELLI benchmark—which primarily uses text adventure games to assess agent behaviorarxiv.org—MoralSim employs formal game structures integrated with moral contexts, allowing for more controlled and easily quantifiable comparisons. This could be better than just simple QA.

**Weaknesses:**

- Scientificness of the evaluation of this assumption. Although multi-agent framework could provide a more vivid setting for revoke LLM's decision under certain scenarios, it could still be a question of how real and how consistent these evaluations are. According to my experience these testings are easily be changed by small parts of prompts. Yet, this paper don't provide a convincing enough evidence to illustrate the scintificness of this testing.

- The novelty is somewhat incremental. Although the MoralSim framework’s integration of morality and game theory is commendable, its concepts overlap with some existing work [1,2,3]. Also, please discuss these papers in the main paper more to let audiance familiar with context and existing research as well as the differences. Especially [2,3] already reported the betray behavior of LLMs

[1] Do the Rewards Justify the Means? Measuring Trade-Offs Between Rewards and Ethical Behavior in the MACHIAVELLI Benchmark

[2] Moral Alignment for LLM Agents

[3] Survival Games: Human-LLM Strategic Showdowns under Severe Resource Scarcity

**Questions:**

- How authors evaluate the testing is stable instead of impacted by small part of prompt design? How often do models change archetype between Goal-oriented vs Neutral prompts? Provide a confusion matrix of archetypes across prompts.

- Please include a comparison table vs MACHIAVELLI, GovSim, Survival Games, and moral-reward alignment detailing the unique “human-harm resource” dimension and survival horizon.

---

> ### Author Response · Authors · 2025-11-20
>
> **Scientificness and Prompt Sensitivity**
> We agree that scientific robustness and sensitivity to prompt wording are critical. In the current paper, we (i) run multiple seeds per configuration, and (ii) conduct a paraphrase robustness study (Sec. 3.5) in which we rephrase the scenarios for two representative configurations and evaluate GPT-4o and DeepSeek-R1. We have now extended this analysis to explicitly quantify the per-paraphrase deltas relative to the baseline absolute values (see Table 1).
>
> For each (model, configuration, paraphrase) combination, we averaged scores over 5 seeds, then computed the difference between each paraphrase and its original version. As shown below, we report the absolute baseline values and the average difference ($\pm$ standard deviation) across paraphrases for each model.
>
> **Table 1: Robustness to Prompt Paraphrasing.** *“Baseline” is the mean value using the original prompt. “$\Delta$” reports the mean difference ($\pm$ standard deviation) between the paraphrased and original prompts across all configurations.*
>
> | Metric | Model | Baseline | $\Delta$ (Mean $\pm$ SD) |
> | :--- | :--- | :---: | :---: |
> | **Cooperation Rate** | DeepSeek-R1 | 50.8% | $+1.1 \pm 1.6$ pp |
> | | GPT-4o | 87.5% | $-3.6 \pm 7.7$ pp |
> | **Self-Payoff** | DeepSeek-R1 | 49.2% | $+0.0 \pm 1.2$ pp |
> | | GPT-4o | 12.1% | $-0.2 \pm 5.3$ pp |
> | **Opponent Alignment** | DeepSeek-R1 | 99.1% | $-1.2 \pm 1.7$ pp |
> | | GPT-4o | 62.7% | $-0.5 \pm 3.7$ pp |
>
> The results indicate high stability for **DeepSeek-R1**, with negligible deviations across all metrics (e.g., cooperation changes by only $1.1 \pm 1.6$ percentage points). **GPT-4o** shows somewhat larger variance (cooperation $\Delta = -3.6 \pm 7.7$ pp), largely driven by a single seed in one configuration that failed to cooperate under a paraphrased prompt. However, even with this outlier, the average shift remains small relative to the baseline.
>
> Across both models, paraphrasing does not flip scenarios from moral to immoral behavior nor reverse the direction of the treatment effects we report. We will include these statistics in the revised paper to make our robustness analysis more explicit.
>
> **Novelty and Comparison to Existing Work**
> We thank the reviewer for highlighting the value of MoralSim as a structured framework for studying LLM behaviour in social dilemmas. Regarding novelty, our goal is complementary to MACHIAVELLI, Moral Alignment for LLM Agents, GovSim, and Survival Games. The following table summarizes the key differences:
>
> **Table 2: Comparison of MoralSim with Related Benchmarks.**
>
> | Aspect | Machiavelli (Pan et al.) | Moral Alignment (Tennant et al.) | GovSim (Piatti et al.) | Survival Games | **MoralSim (ours)** |
> | :--- | :--- | :--- | :--- | :--- | :--- |
> | **Environment / game structure** | Narrative CYOA; non-canonical dilemmas | Canonical matrix games (IPD + variants) | Commons-resource multi-agent sim; non-canonical | Rich survival-town sim; non-canonical | **Canonical repeated PD + PGG social dilemmas** |
> | **Explicit moral vs payoff conflict** | $\checkmark$ (harm vs reward) | $\checkmark$ (intrinsic moral rewards) | ~ (sustainability vs exploitation) | $\checkmark$ (survival vs ethics) | **$\checkmark$ (per-round cooperation vs payoff)** |
> | **Human-harm / societal impact dimension** | $\checkmark$ (dense harm / wellbeing labels) | ~ (moral axioms, no explicit harm labels) | ~ (commons overuse harms group) | $\checkmark$ (harm, trust, exploitation) | **$\checkmark$ (moral framings: contracts, privacy, climate)** |
> | **Resource / survival horizon** | $\times$ (no explicit resource dynamics) | $\times$ | $\checkmark$ (resource sustainability over time) | $\checkmark$ (agent survival) | **$\checkmark$** |
> | **Factorial design across game, context, opponent, risk** | $\times$ | $\times$ | $\times$ | $\times$ | **$\checkmark$** |
> | **Causal analysis over factors** | $\times$ | $\times$ | $\times$ | $\times$ | **$\checkmark$** |
> | **Evaluates multiple frontier models zero-shot** | $\checkmark$ | $\times$ (focus on one 2B model + fine-tuning) | $\checkmark$ | $\checkmark$ | **$\checkmark$** |
>
> Our main conceptual contribution is to show that, even in this controlled setting, no frontier model behaves morally in a context-robust way: their choices flip between moral and payoff-maximizing regimes as we vary game, moral domain, opponent, and survival risk.

---

### Meta-Review · Area_Chair_NyKp · 2026-01-06

**Summary:**

The reviewers reached a mixed consensus (Ratings: 2, 4, 6, 6), with the following primary concerns:

- Limited Technical Novelty and Contribution: Reviewers questioned the incremental nature of the work compared to existing benchmarks like MACHIAVELLI, GovSim, and particularly Tennant et al. (2024/2025), which also explores moral alignment in the Prisoner’s Dilemma.
- Prompt Sensitivity and Robustness: Multiple reviewers were concerned that the observed behaviors might be artifacts of specific prompt phrasing rather than true "moral policies".
- Realism and Evaluation-Awareness: Reviewer raised the possibility of "scheming," where frontier models might recognize they are in an evaluation environment and alter their behavior accordingly.
- Scope Limitations: Reviewer noted the narrow focus on two-player canonical games (PD and PGG) without allowing for multi-agent social dynamics or negotiation.
- Procedural Violation (Page Limits): Reviewer w88c pointed out that moving the Related Work section to the Appendix appeared to be a method for bypassing the 10-page main body limit, which was deemed unfair to other authors.

**Reviewer Concerns:**

Addressed Concerns:

• Prompt Sensitivity: The authors expanded their robustness study, showing that paraphrasing only caused a ~1.1–3.6 percentage point shift in metrics, proving the findings are not mere artifacts of specific wording.
- Comparison to Existing Benchmarks: The authors provided a comprehensive comparison table (Table 2 in the rebuttal), distinguishing MORALSIM by its factorial design and explicit causal analysis of marginal effects.
- Evaluation-Awareness: Using an LLM judge, the authors demonstrated that only 2.7% of runs showed explicit "evaluation-awareness," primarily concentrated in the DeepSeek-R1 model.
- Temporal Dynamics: The authors added an "early vs. late" cooperation analysis to address the lack of temporal depth, finding evidence of free-riding decay in public goods games.

Outstanding Concerns
- Procedural Integrity: The Related Work section remains in the Appendix.
- Conceptual Overlap: While the authors clarified the difference between MORALSIM (an evaluation benchmark) and Tennant et al. (a training method), the underlying game structures are very similar, and the reviewer (w88c) remained unconvinced of the "substantial contribution".
- Simulation vs. Reality: The limitation to non-communicating two-player games remains an abstraction that reviewers felt might not capture the nuances of actual moral deployment.

**Reviewer Scores:**

-Reviewer VVKc (Initial 4): Projected 5. The extensive robustness data and the comparison table directly addressed their primary requests. They likely move to a marginal accept.
- Reviewer 8FDD (Initial 6):  No Change.. This reviewer was already leaning positive. The "scheming" analysis was helpful, but the fundamental concern about the "realistic" text of the games remains.
- Reviewer mAm7 (Initial 6): No Change. This reviewer explicitly stated the rebuttal addressed their concerns and chose to retain their score.
• Reviewer w88c (Initial 2): Projected 3. While the authors' clarification on Tennant et al. was logical, this reviewer was highly critical of the page-limit violation and the marginal nature of the contribution, making a move to the acceptance side unlikely.

---

### Decision · Program_Chairs · 2026-01-26

Reject